# Safety and Immunogenicity of the Convacell^®^ Recombinant N Protein COVID-19 Vaccine

**DOI:** 10.3390/vaccines12010100

**Published:** 2024-01-19

**Authors:** Sevastyan Rabdano, Ellina Ruzanova, Denis Makarov, Anastasiya Vertyachikh, Valeriya Teplykh, German Rudakov, Iuliia Pletyukhina, Nikita Saveliev, Konstantin Zakharov, Diana Alpenidze, Vasiliy Vasilyuk, Sergei Arakelov, Veronika Skvortsova

**Affiliations:** 1Saint Petersburg Scientific Research Institute of Vaccines and Serums of the Federal Medical-Biological Agency of Russia (SPbSRIVS), St. Petersburg 198320, Russiay.v.pletuhina@niivs.ru (I.P.);; 2LLC “NIC Eco-bezopasnost”, St. Petersburg 191119, Russia; 3State Budgetary Health Institution “City Polyclinic No. 117”, St. Petersburg 194358, Russia; 4Department of Toxicology, Extreme and Diving Medicine, North-Western State Medical University named after I.I. Mechnikov, St. Petersburg 191015, Russia; 5Federal Medical-Biological Agency of Russia, Moscow 115682, Russia

**Keywords:** COVID-19, vaccine, recombinant, clinical trial, subunit, nucleocapsid

## Abstract

We have developed Convacell^®^—a COVID-19 vaccine based on the recombinant nucleocapsid (N) protein of SARS-CoV-2. This paper details Convacell’s^®^ combined phase I/II and IIb randomized, double-blind, interventional clinical trials. The primary endpoints were the frequency of adverse effects (AEs) and the titers of specific anti-N IgGs induced by the vaccination; secondary endpoints included the nature of the immune response. Convacell^®^ demonstrated high safety in phase I with no severe AEs detected, 100% seroconversion by day 42 and high and sustained for 350 days anti-N IgG levels in phase II. Convacell^®^ also demonstrated a fused cellular and humoral immune response. Phase IIb results showed significant post-vaccination increases in circulating anti-N IgG and N protein-specific IFNγ^+^-producing PBMC quantities among 438 volunteers. Convacell^®^ showed same level of immunological efficacy for single and double dose vaccination regimens, including for elderly patients. The clinical studies indicate that Convacell^®^ is safe and highly immunogenic.

## 1. Introduction

Post-pandemic, COVID-19 vaccines remain the best-developed and most widespread treatment strategy for the disease [1], able to both protect the vulnerable groups and lessen the disease’s impact on the populace and workforce [2,3,4].

To diversify the current armamentarium of COVID-19 vaccines with alternative antigens options, we have developed our vaccine Convacell^®^ [5,6]. Convacell^®^ is based on the full-length nucleocapsid (N) protein of SARS-CoV-2, produced in an *Escherichia coli* recombinant protein platform. Recombinant protein vaccines in general are safe and well tolerated, and this platform in particular is proven to produce vaccines with good safety profiles, due to high antigen quality and a lack of ballast impurities [7,8]. A dose of Convacell^®^ contains 50 μg of protein N with the Wuhan SARS-CoV-2 variant sequence. The N protein is highly conserved and less prone to accumulate mutations than the commonly used in vaccines spike (S) protein [9,10,11,12,13,14], which allows Convacell^®^ to generate long-lasting [5] and broadly acting [15] immune responses. The N protein is also abundantly expressed in cells after infection [16,17] and exposed on the infected cells membranes [18,19], ensuring that infected cells are highly likely to be targeted by N-specific immune responses. Such responses then rapidly clear the infected cells via cytotoxic T-cell [20,21,22] and natural killer cell action [23,24,25]. COVID-19 vaccines based on the N protein have already been described as promising in multiple papers [9,10,26,27,28,29,30]. Studies have shown that immunization with the N protein induces the creation of tissue memory cells (T_RM_ cells) in lungs [31], that it protects from severe disease [29,32], and that the immune response acquired this way is long-lasting [10,33]. Supporting these findings are the results of immunizations with the nucleocapsid proteins of other viruses, which show similar effects in mice [34,35,36]. Preclinical studies of Convacell^®^ indicated that the vaccine is safe and effective at preventing severe disease [5]. 

Convacell^®^ differs from currently broadly used spike protein-based vaccines, both in the target antigen and in the dominant mechanism of immune response. Hence, it is likely to be useful for vaccinating so-called “non-responders” [37,38,39,40,41,42], who cannot form an S-specific immune response. Moreover, the longevity of the immune response generated by Convacell^®^ [5] indicates that Convacell^®^ is likely to not require booster doses to achieve maximum effectiveness and that vaccination with Convacell^®^ is likely to be organizationally simple.

In this study, we describe the results of the phases I, II and IIb of Convacell^®^’s clinical trials, which include assessments of safety and immunogenicity of Convacell^®^, as well as determination of its optimal immunization regimen and its suitability for the elderly.

## 2. Materials and Methods

### 2.1. Vaccine Formulation

The volunteers in the study received one or two doses of a recombinant subunit COVID-19 vaccine based on the nucleocapsid protein of SARS-CoV-2, Convacell^®^, in the form of emulsion for intramuscular injection. Convacell^®^ has been examined in preclinical trials and found to be both safe and effective at inducing an immune response [5].

Each 0.5 mL dose of Convacell^®^ contains 50 µg of recombinant SARS-CoV-2 nucleocapsid protein as the main active ingredient. Supplementary ingredients are 5 mg of (±)-α-tocopherol, 15 mg of squalane and 5 mg of polysorbate 80 in form of nano-emulsion.

The vaccine used in this study was produced according to the GMP by the Saint Petersburg Scientific Research Institute of Vaccines and Serums (SPbSRIVS). Lot #460621R was used in phase I, lot #470621R was used in phase II, and lot #40822 was used for phase IIb.

The placebo formulation used in the study was identical to the vaccine formulation, with the exception of containing no SARS-CoV-2 nucleocapsid protein.

### 2.2. Study Design

The phase I/II study (NCT05156723) combined two phases in one protocol, with phase I being open-label and phase II being randomized and double-blind. In phase I, the study recruited 20 volunteers total, divided into two groups: group 1 (*n* = 5) and group 2 (*n* = 15). Volunteers in both groups were vaccinated once with the studied vaccine formulation. Vaccinated volunteers were hospitalized for observation over 7 days and then observed further in ambulatory conditions for a total of 21 days. Group 1 assessed vaccine safety over 7 days; after no severe adverse effects (SAEs) were observed in group 1, group 2 was recruited and vaccinated to assess vaccine safety in more detail. 

After the successful assessment of vaccine safety, the study advanced onto phase II, recruiting 135 total volunteers, divided into 3 groups numbered 3–5 (*n* = 45 in each). Vaccinations/placebo injections were performed on days 0 and 21: two vaccinations for group 3, one vaccination and one placebo injection for group 4, and two placebo injections for group 5. The volunteers in phase II were observed in ambulatory conditions for a total of 350 days to assess safety and immunogenicity of the vaccine.

During the course of the phase II study, two cohorts of volunteers were selected from the total population for in-depth studies: cohort A (*n* = 15 per group), to study the specific, especially cellular, nature of post-vaccination immune response and the mechanism of action of the vaccine on the immune system; and cohort B (*n* = 15 per group), to assess the long-term immunogenicity of the vaccine, on days 240 and 350 after first vaccination.

The schematic representation of the study design is depicted in Figure 1.

The phase IIb study (NCT05726084) initially aimed to determine the optimal vaccination regimen (1 or 2 doses) for Convacell^®^, based on the results obtained in phase II. The study was randomized, double-blind, and involved volunteers with no upper age limit. It followed the same general design as the phase II study: after successful screening, the volunteers were randomly divided into two groups. Group 1 (*n* = 215) received vaccine on day 0 and placebo on day 21. Group 2 (*n* = 218) received vaccine on days 0 and 21. Blood sample collection was conducted on day 42 for both groups. The schematic representation of the study design is depicted in Figure 2.

### 2.3. Eligibility Criteria

Both studies recruited volunteers meeting all of the following eligibility criteria:
Age 18 to 60, or no upper age limit for study IIb;Willing to sign an informed consent statement to participate in a clinical trial;18.5 ≤ BMI ≤ 30 kg/m^2^, with body mass between 55 and 100 kg for men and between 45 and 100 kg for women;Verified healthy status: no deviation from reference intervals in the results of standard clinical and laboratory tests;Negative for: human immunodeficiency virus (HIV), rapid plasma reagin (RPR), hepatitis B surface antigen (HBsAg), hepatitis C virus RNA (HCVRNA);Hemodynamic and vital parameters within following reference intervals: heart rate 60–90 bpm, respiratory rate under 22 breaths per minute, systolic arterial pressure 100–139 mmHg, diastolic arterial pressure 60–89 mmHg;Willing to keep a self-observation diary and attend control visits;Willing to abstain from alcohol for 14 days before the beginning of the study and until its completion;Willing to abstain from smoking for 48 h before the beginning of the study and while hospitalized;For fertile women: negative pregnancy test and willing to use adequate contraception methods until the completion of the study and for at least two months after vaccination;For fertile men: willing to use adequate contraception methods until the completion of the study or past vasectomy with confirmed azoospermia, partner willing to use at least 90% effective contraception methods or past tubal ligation or menopausal for at least 2 years.

### 2.4. Sample Size Determination

In phase I, no formal sample size calculation was made, as no hypotheses were tested.

In phases II and IIb, formal sample size calculations were carried out via a validated copy of PASS 2021 software, version 21.0.3 (NCSS Statistical Software, Kaysville, UT, USA), itself based on the works of O’Hagan, Stevens, and Campbell in sample size statistics [43]. 

In phase II, the mode used was one-tailed superiority trial, with no threshold for non-marginal superiority, for each pair of values of vaccinated groups (i.e., 3 and 4) compared to the placebo group (i.e., 5). Desired type I (false-positive) error rate was set to 0.0125, and desired type II (false negative) error rate was set to 0.2. The results indicated that the minimum sample size for each group to achieve the desired results would be 41. With arbitrarily assumed dropout allowance of 9%, it was planned to screen 45 volunteers per group.

In phase IIb, the mode used was a *t*-test of two geometric means of a log-normal distributed variable for two groups. Desired type I (false-positive) error rate was set to 0.025, and desired type II (false negative) error rate was set to 0.1. A dropout rate of 10% was formally factored in. The results indicated that the minimum sample size for each group to achieve the desired results would be 230 with dropouts.

### 2.5. Ethical Committee

The combined phase I/II study protocol was approved by the Ethics Committee of the Ministry of Health of the Russian Federation (decision #388 from 19 July 2021). The involvement of additional research centers in this study was additionally approved by the same body (decision #4183295-20-1/ДP from 18 August 2021).

The phase IIb study protocol was approved by the Ethics Committee of the Ministry of Health of the Russian Federation (decision #583 from 5 October 2022). The involvement of additional research centers in this study was additionally approved by the same body (decision #4230567-25-2/ДP from 14 October 2022). 

The independent ethical committees of all involved research center have also approved both aforementioned protocols (decision data available upon request).

### 2.6. Outcomes and Assessment

The primary safety endpoint of phases I and II was the frequency of local and systemic adverse effects (AEs) observed in the span of 21 days after the first vaccination. AEs were determined as disturbances in the tested vital parameters; or disorders that arose during the course of the study and were detected as the result of assessment by a professional physician. Physician assessments and sample collection for vital parameter assessment were performed at each visit. All AEs that were observed during the course of the safety study were recorded, regardless of their putative association with the administration of the studied vaccine formulation. Severe AEs (SAEs) were defined as any AE that led to the hospitalization of the volunteer and/or required immediate medical intervention, and/or led to the volunteer’s death.

The primary immunogenicity endpoint, used for phases II and IIb, was the mean of the quantity of specific anti-N antibodies in the sera of vaccinated volunteers on days 21 (only for phase II) and 42 after first vaccination. Also studied was the quantity of specific anti-N antibodies in the sera of vaccinated individuals on days 14, 21, 28, 42, 90, and 180. For cohort B in phase II, time points 240 and 350 days after vaccination were also investigated. The frequency of seroconversion, defined as the presence of any specific anti-N antibodies in sera, and the nature of specific post-vaccination immune response in volunteers at each time point were studied on same study days as for specific anti-N antibodies.

To obtain volunteer sera and PBMC, volunteers’ blood was collected into 6 mL vacuum tubes containing K2 EDTA as anticoagulant. The blood was then separated into sera and PBMC using centrifugation in a ficoll density gradient.

The quantities of specific anti-N IgG antibodies in volunteer sera were assessed via the AdviseDx SARS-CoV-2 IgG II chemiluminescent microparticle immunoassay (Abbott Laboratories, Chicago, IL, USA). The quantities of specific anti-N IgM antibodies in sera were assessed via the SARS-CoV-2-IgM-IFA-BEST ELISA kit (Vektor BEST, Novosibirsk, Russia D-52). The standard manufacturer protocol was followed in each case.

Examination of the nature of specific post-vaccination immune response in cohort A volunteers involved the assessment of antibody-dependent natural killer cell activation (ADNKA) elicited by the anti-N immunoglobulins obtained from the volunteers’ sera; that is, the quantification of the ability of vaccination-generated anti-N antibodies to drive natural killer cell activation upon encountering the N protein [44,45,46]. The method used here, which analyzes CD107a expression on the surface of NK cells as a marker of degranulation and stimulation, was adapted from Fielding et al. [19]. In the beginning, 100 µL of heat-inactivated volunteer sera were added to the wells of a 96-wells microplate containing immobilized SARS-CoV-2 N-protein. After 2 h incubation at 37 °C, 3 × 10^5^ PBMC in 100 µL of AIM-V media (Invitrogen, Carlsbad, CA, USA) was added to each well, followed by fluorescently tagged antibodies against CD107a. The resulting mixture was then incubated for 2 h at 37 °C and 5% CO_2_. Then, 10 µg/mL Brefeldin A was added to the cell mixture and incubation was continued for another 2 h at 37 °C and 5% CO_2_. After the incubation, the cells were washed in Dulbecco’s phosphate-buffered saline (DPBS, Sigma-Aldrich, Burlington, MA, USA), and mixed with fluorescently labelled antibodies against surface cytokines CD56, CD16, CD3 and 7-AAD. The cells were then fixed via paraformaldehyde and analyzed via the CytoFlex Flow Cytometer (Beckman Coulter, Chaska, MN, USA). Results were analyzed via the CytExpert Acquisition and Analysis Software V.2.4 (Beckman Coulter, USA). ADNKA levels were assumed to be directly proportional to the numbers of CD3^−^CD16^+^CD56^+^7-AAD^−^CD107a^+^ cells, i.e., degranulated NK cells expressing CD107a.

To further assess the nature of specific post-vaccination immune response in cohort A, quantification and phenotyping of the N-specific T-cells in volunteer blood samples was performed via flow cytometry after their stimulation with a SARS-CoV-2 N-protein peptide pool. To achieve this, 10^6^ volunteer PBMC in 100 µL culture media were supplemented with 1 µg/mL pooled N peptides—Peptivator (Miltenyi biotec, Bergisch Gladbach, Germany)—and incubated for 12 h at 37 °C and 5% CO_2_. Two hours after the addition of pooled peptides, 10 µg/mL Brefeldin A (Sigma-Aldrich, USA) was added. After incubation, the cells were washed in DPBS and stained for surface-expressed cytokines via antibody conjugates anti-CD3 (UCHT1), anti-CD4 (13B8.2), anti-CD8 (B9.11), anti-CD45RA (2H4), anti-CD197 (G043H7) (all Beckman Coulter, USA). PBMC viability was assessed via the Zombie Aqua™ Fixable Viability Kit (Biolegend, San Diego, CA, USA). Stained cells were washed in DPBS (Sigma-Aldrich, Burlington, MA, USA), fixed and permeabilized via the IntraPrep permeabilization Reagent (Beckman Coulter, Chaska, MN, USA). Finally, the permeabilized cells were stained against internal cytokines IL-2 and IFNγ via antibody conjugates anti-IL-2 (IL2.39.1) and anti-IFNg (45.15) (all Beckman Coulter, Chaska, MN, USA) The stained cells were analyzed via the CytoFlex Flow Cytometer (Beckman Coulter, Chaska, MN, USA), with each sample registering no less than 10,000 hits. Results, specifically the CD4^+^IFNγ^+^, CD4^+^IL-2^+^, CD8^+^IFNγ^+^, and CD4^+^IL-2^+^ cell populations, were analyzed via the CytExpert Acquisition and Analysis Software V.2.4 (Beckman Coulter, Chaska, MN, USA).

The nature of specific post-vaccination immunity was assessed in phase IIb via using the TigraTest^®^ SARS-CoV-2 ELISPOT kit (Generium, Moscow, Russia) to count the IFNγ-producing PBMCs volunteers’ samples. The kit does not specify the precise nature of its probes; however, the authors have previously published a paper detailing the probes used in the kit, which all appear to bae taken from conservative virus genes [47,48]. Cryovials with volunteers’ PBMCs were thawed in a 37 °C water bath, washed in 10 mL of the kit’s warmed culture media, counted in a Goryaev chamber and assessed for viability via the Zombie Aqua™ Fixable Viability Kit (Biolegend, San Diego, CA, USA). Samples with >90% viability were selected for future analysis. Selected samples were diluted to 10^6^/mL concentration in AIM-V (Invitrogen, USA) and incubated for 8–12 h at 37 °C and 5% CO_2_. Then, 100 µL of each sample were added to a well in the kit’s 96-well ELISPOT membrane microplate, and supplemented with 100 µL, 0.6 nM/mL either of peptide pool 1 (S-protein) or peptide pool 2 (N-, membrane and envelope proteins of SARS-CoV-2) in AIM-V (Invitrogen, Carlsbad, CA, USA). The microplate was then incubated for 18–24 h at 37 °C and 5% CO_2_, washed six times, supplemented with the kit’s horseradish peroxidase-conjugated anti-IFNγ IgG and incubated for 2 h at room temperature. After incubation, the plate was washed 6 times, supplemented with 100 µL of the kit’s horseradish peroxidase substrate chromogen into each well, and incubated for 20 min in darkness at room temperature. Then, the reaction was stopped through washing the microplate with deionized H_2_O and drying it at 37 °C in a thermostat. The spots on the membrane were counted via the iSpot microplate reader (AID Autoimmun Diagnostika GmbH, Strassberg, Germany).

### 2.7. Statistical Analysis

Statistical analysis of the data presented in tables in this paper was performed via the R statistical package, version 4.0.2, adapted and validated by Microsoft (Microsoft R Open, Redmond, WA, USA), with the usage of “A Guidance Document for the Use of R in Regulated Clinical Trial Environments” [49].

Statistical analysis of the data presented in graphs in this paper was performed independently by the papers’ authors based on the raw data obtained in the study. The analysis was performed in the GraphPad Prism software for version 9.5.0. (GraphPad, Boston, MA, USA). The statistical tests used were Student’s *t*-tests for comparison between two samples and ANOVAs for comparison between multiple samples, the specific types of ANOVA used were as indicated in the results section.

## 3. Results

### 3.1. Patients

The combined phase I/II trial lasted from 29 July 2021 until 12 October 2022 and screened 30 healthy volunteers in phase I and 170 healthy volunteers in phase II, of which 20 for phase I and 134 for phase II were subsequently involved in the study and randomly assigned into groups. The screening and randomization for phase II was performed in four centers, as described in Table 1:

Volunteer distribution divided into immunogenicity assessment and safety assessment cohorts, and the number of dropouts by category in phase II per group is detailed in Table 2. The first 15 volunteers of each group enrolled in Center 2 were included into cohorts A and B. One additional volunteer was included into cohort B from group 3.

An overview of patient demographic data for both phases are presented in Appendix A. Note that while the mean weight of volunteers did differ significantly between groups in phase II, the BMI did not.

The phase IIb trial lasted from 24 October 2022 until 19 December 2022 and screened 470 volunteers, of whom 433 were randomized. Nine dropouts in total were registered. An overview of volunteer distribution by group and the number of dropouts by category is presented in Table 3. An overview of patients’ demographic data are presented in Appendix A.

### 3.2. Safety

Phase I revealed (Table 4) that the vaccine is safe and well tolerated, as the related AEs observed were overwhelmingly mild. No SAEs were observed at any point. The only categories of related AEs observed were either localized reactions at the injection site or disturbances of investigated parameters. Detailed data for total (related and unrelated) AEs in Phase I are presented in Appendix A. Data includes AEs observed after day 21 until the end of the experiment according to phase I protocol.

Phase II results (Table 5) exhibited a pattern of predominantly mild related AEs, with local reactions at the injection site being the most common. No SAEs were observed. Detailed data for total (related and unrelated) AEs in phase II are presented in Appendix A. Phase IIb results (Table 6) continued the pattern of predominantly mild local reactions to vaccination. No SAEs were observed. Detailed data for total (related and unrelated) AEs in phase IIb are presented in Appendix A.

In phase II, one case of independently confirmed via PCR COVID-19 was observed in the single vaccinated group, two cases were in the double vaccinated group, and five cases were in the placebo group. Notably, the one PCR-confirmed case in the single vaccination group occurred during the first week after vaccination. In phase IIb, one case of independently confirmed via PCR COVID-19 was observed in the double vaccinated group.

### 3.3. Humoral Immune Response

The assessment of circulating IgG levels against the N protein of SARS-CoV-2 in phase I volunteers (Appendix A) revealed a statistically significant (*p* < 0.0001) increase in detected anti-N IgG quantity in sera on day 21 compared to day 0. The assessment of circulating IgM levels against the N protein in phase I volunteers (Appendix A) revealed no statistically significant increase (*p* = 0.08) in specific antibody quantities on day 21 compared to day 0.

In phase II, vaccination generated high circulating anti-N IgG levels, which lasted up to 350 days (Figure 3). From day 14 up to day 180, vaccinated individuals had significantly higher quantities of circulating anti-N IgG compared to the placebo; however, after day 180, this difference disappeared due to the placebo group’s circulating anti-N titers rising. The differences between single vaccination and double vaccination cohorts were minor and not statistically significant. Seroconversion was at 100% on day 42 in both vaccinated groups. 

Graphing individual values for circulating anti-N IgG quantities in sera (Appendix A) revealed both the single vaccination and the double vaccination groups noticeably separate into two subgroups: one demonstrated a rapid response, with a major increase in circulating anti-N IgG quantities on day 14 post-vaccination, while the second exhibited a slower response, with a major increase in circulating anti-N IgG quantities only on day 28 post-vaccination. This separation could be explained by entry-point anti-N IgG levels (Appendix A). Analysis of the circulating anti-N IgG quantities on day 0 vs. circulating anti-N IgG quantities on day 14 for the single and double vaccination groups revealed a positive correlation between the two values, with the *p*-values 0.004 and 0.01 for the single and double vaccination groups, respectively.

Assessing the circulating anti-N IgM quantities in vaccinated and placebo-injected individuals (Appendix A) revealed a tendency for both vaccinated groups to have higher anti-N IgM titers in sera from day 14 until convergence on day 90, though only on day 28 did the vaccinated groups have significantly higher circulating anti-N IgM quantities compared to the placebo group.

### 3.4. Cellular Immune Response

The ability of Convacell^®^ to induce cellular immune responses was assessed in cohort A (*n* = 45) in phase II via antibody-dependent natural killer cell activation (ADNKA) and T-cell activation by the N protein. Anti-N IgG can form immune complexes with N, which are then capable of activating natural killer (NK) cells. Activated NK cells then release cytotoxic granules via a process called degranulation, which is associated with the appearance of CD107a molecules on the surface of the cell [19]. CD107a can therefore be used as a cytometric marker of NK cell activation, which is how NK activation was assayed by us. Analysis via Mann–Whitney tests of natural killer cell activation by N-IgG immune complexes in volunteer sera (Figure 4) revealed statistically significant increases in the percentages of activated NK cells on experiment days 42 and 240 for the single vaccination group and on days 42–350 for the double vaccination group, compared to the placebo group. No significant differences were observed between the two vaccinated groups.

We also assessed the specific activation of volunteer T-cells by the N-protein. Median values over time are presented in Appendix A, individual values over time are presented in Appendix A. Statistical comparison within groups over time via Wilcoxon signed-rank test revealed a statistically significant increase in the single vaccination cohort in CD4^+^IFNγ^+^ percentage on experiment days 180, 240, and 350 compared to experiment day 0. Also revealed was a significant decrease in CD8^+^IFNγ^+^ percentage compared to day 0 on day 180 for the placebo and double vaccination groups. Finally, for the placebo group, there was a significant spike in CD8^+^IL-2^+^ percentage on day 180 compared to day 0. Statistical comparison between groups via Mann–Whitney tests revealed no significant differences between the groups at any time point.

### 3.5. Single vs. Double Vaccination

Both groups demonstrated significant increases from day 0 to day 42 in circulating anti-N IgG quantities (Figure 5A). An assessment of the effect of immunization regimen on circulating specific anti-N IgG quantities in phase IIb revealed no statistically significant differences between single and double vaccination groups on day 42. Both groups demonstrated significant increases from day 0 to day 42 in specific IFNγ-producing PBMC quantities for cells stimulated with peptide pool containing protein N epitopes (Figure 5B). An assessment of the effect of the immunization regimen on the numbers of circulating N-specific IFNγ^+^ T-cells came to the same conclusion—there were no statistically significant differences between single and double vaccination groups on day 42, though the double vaccination group did have a trend towards a higher quantity of IFNγ-producing T-cells. An analysis of only the data of volunteers aged 60+ (Appendix A) revealed a similar pattern of no significant differences between two groups on day 42 for either anti-N IgG or IFNγ-producing T-cells. Both elderly groups demonstrated significant increases from day 0 to day 42 in circulating anti-N IgG quantities and IFNγ-producing PBMC quantities.

Assessing circulating anti-S IFNγ^+^ PBMC levels in the blood of volunteers on day 0 (Appendix A) demonstrated low levels of activated PBMCs across both groups, and no significant difference between groups. Therefore, the data indicated that the volunteers were predominantly SARS-CoV-2-naïve.

### 3.6. Infections and Virus Encounters

Although assessing the statistical effectiveness of Convacell^®^ was not the endpoint of the study phases presented in this paper, the PCR-confirmed COVID-19 incidence data among the various groups in phase II nevertheless reveals a pattern with possible implications about Convacell^®^’s ability to protect against COVID-19 (Figure 6A). Vaccinated groups showed fewer PCR-confirmed COVID-19 cases than the placebo group. However, we cannot yet conclude that Convacell^®^ vaccination protects from COVID-19 based on this data alone, as the sample size is too small to allow for statistically significant results.

Encounters with the SARS-CoV-2 do not necessarily lead to COVID-19; in many cases, SARS-CoV-2 infection is asymptomatic [50,51,52]. However, such asymptomatic encounters can be detected by serological tests that analyze the quantity of specific antibodies against the virus [53]. In this way, by further analyzing the data, it is also possible to approximately determine the number of putative asymptomatic SARS-CoV-2 infections in our study. As overall anti-N IgG quantities in both vaccinated groups peaked on day 42 post-vaccination, it can be assumed that any increase past a certain threshold in anti-N IgG levels in any individual past this time point is likely due to an infection. We have chosen the value of the threshold to be 1.4 RLU, as it is the minimal value considered to be positive according to the assay manual. By counting and plotting the number of volunteers in each group who have experienced at least one instance of increase beyond the of 1.4 RLU in anti-N IgG quantities past experiment day 42, it is possible to create a graph (raw anti-N IgG quantification data is supplied in SI in spreadsheet format) of putative asymptomatic SARS-CoV-2 infections past day 42 among all groups, as depicted in Figure 6B.

Putative asymptomatic infections in the placebo group were identified in 88.6% of the placebo group volunteers. Vaccination led to a decrease in putative asymptomatic infection incidence to either 50%, for the single vaccination group, or 59.1%, for the double vaccination group. Taking into account the assumption that all of the subjects in the study were in the same environmental conditions, the probability of a virus encounter was the same across the groups. The observed drop in the number of putative asymptomatic infections for vaccinated groups therefore suggests that Convacell protects not only from COVID-19, but also from SARS-CoV-2 infections.

## 4. Discussion

The safety assessment of Convacell^®^ in phases I, II, and IIb has shown that no severe AEs occurred at all, and the overall pattern of AEs was dominated by mild localized reactions and mild systemic effects; e.g., injection site pain and headache were the most frequent AE types observed. In both the single and double vaccination groups, the overall pattern of mild localized and transient systemic AEs was similar to that in the placebo group. Overall, the results indicate that Convacell^®^ administration carries with it no significant or long-term health risks, and can be considered to be safe for healthy adults.

The main endpoint of the assessment of Convacell^®^ immunogenicity, the values of specific anti-N antibody quantities on days 21 and 42, has demonstrated that both single and double vaccinations with Convacell^®^ result in great increases in specific anti-N IgG levels. Both in phase I and phase II, specific anti-N IgG quantities in the sera of vaccinated volunteers, compared to the placebo group, were higher on day 21. On day 42, in phase II, all vaccinated volunteers achieved full seroconversion and uniformly high circulating specific anti-N IgG levels. Past day 42, circulating anti-N IgG for both groups slowly declined, but were maintained at a high level, regardless, even after they plateaued on day 240. In phase IIb, the results of the immunogenicity assessment were especially notable, due to the large sample sizes—these results indicate that Convacell^®^ is highly immunogenic, even with a single dose vaccination regimen, as N-specific IgG quantities significantly increased in the blood of volunteers of both groups 1 and 2 from day 0 to day 42. The low levels of S-specific IFNγ-producing PBMC levels observed in volunteers in phase IIb on day 0 indicate that those volunteers were predominantly SARS-CoV-2 naïve, and that it was vaccination and not infection that led to the increase in anti-N IgG quantities over time observed in that phase.

In contrast to IgG, specific anti-N IgM quantities do not appear to be majorly influenced by vaccination with Convacell^®^.

The phenotyping of activated PBMCs in response to stimulation with the N-protein-derived peptides in the peripheral blood of vaccinated volunteers in phase II revealed a statistically significant increase from approximately 0.5 activated cells per 10,000 to approximately 2 activated cells per 10,000, compared to time point 0, only for the CD4^+^IFNγ^+^ T-cells of the single vaccination group, on days 180–350. This is indicative of a weaker response than that observed in the results of Convacell’s ^®^ preclinical evaluations, where we confirmed that Convacell^®^ induces strong cellular immune response in non-human primates and mice [5,15]. The phase IIb findings indicated a significant and rapid rise from day 0 to day 42 in the levels of circulating N-specific IFNγ-producing PBMCs in both single vaccination and double vaccination groups, regardless of volunteer age. The peptides used to stimulate PBMCs in phase IIb contained the viral envelope and membrane proteins, in addition to the N-protein; however, the observed immune response was N-specific, due to the vaccine that induced it containing only the N-protein.

Assessing the activation of NK cells by the N-IgG immune complexes of vaccinated volunteers revealed that both vaccinated groups exhibited a significant increase in the percentage of activated cells on and after day 42, compared to the placebo group on the same day. This indicates that Convacell^®^ vaccination increases N-specific NK cell activation in the recipient, which is reportedly a major component of a natural anti-SARS-CoV-2 immune response [19,24,54].

Notably, at no point did the single and double vaccination groups, in either phase II or IIb, exhibit any significant differences between their metrics of immunogenicity, which allows us to conclude that a single dose regimen of Convacell^®^ performs identically, with regard to the generation of the immune response, to the two-dose regimen, and that for Convacell^®^ to perform optimally, only a single vaccination is required.

While no formal assessment of Convacell^®^‘s protective effect against COVID-19 was planned in phases I, II, and IIb, the available group-separated COVID-19 incidence data in phase II suggests that the vaccine has a protective effect nevertheless: the placebo group had more disease cases than each of the vaccinated groups. This observation is reinforced by the fact that phase II took place during a spike in COVID-19 cases in Russia (Appendix A). Moreover, inferring the number of putative asymptomatic infections from IgG quantities data reveals that each of the vaccinated groups had roughly half of the putative asymptomatic infections past experiment day 42 than the placebo group had. This observation in turn suggests that the vaccine provides protection from SARS-CoV-2 infections, although an admission needs to be made that on day 240 and 350, we only had data for cohort B (*n* = 46), which might have lowered the reliability of our findings for those two time points. Notably, considering that the main circulating variant of SARS-CoV-2 during the course of the study was Omicron, and that Convacell^®^ is based on the N protein of the Wuhan variant, the conclusions above can be extended to suggest that Convacell^®^ generates an immune response that is cross-specific against the Omicron and the Wuhan variants of the virus. This is fully congruous with the results of our earlier study [15]. Phase IIb contained no placebo control groups, and therefore no conclusion can be reached about the putative effectiveness of Convacell^®^ based on the COVID incidence observed in that phase.

In summary, the results of this trial provide evidence that Convacell^®^ is safe, able to generate a strong and long-lasting humoral immune response, and requires only a single dose to be effective. The findings of this study indicate that Convacell^®^ is likely to be a useful addition to the existing armamentarium of COVID-19 vaccines. Phase III clinical trials are consequently currently underway (id: NCT05726084 on clinicaltrials.gov).

## Figures and Tables

**Figure 1 vaccines-12-00100-f001:**
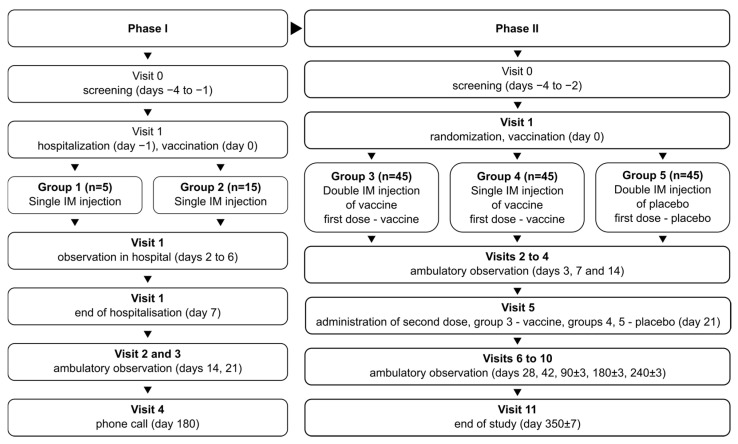
Schematic representation of the study design for phases I and II.

**Figure 2 vaccines-12-00100-f002:**
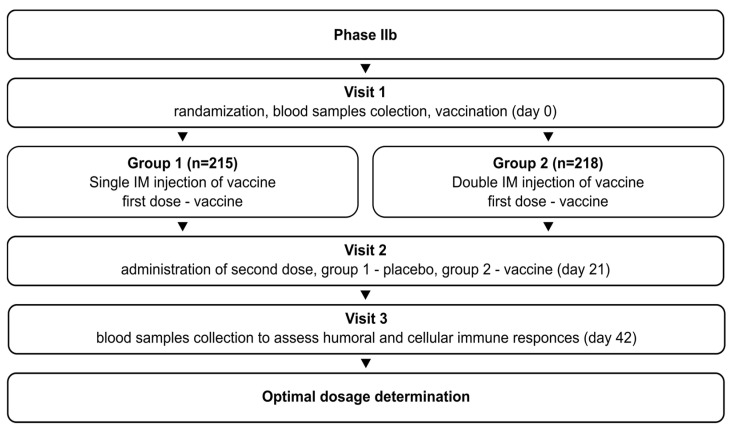
Schematic representation of the study design for phase IIb.

**Figure 3 vaccines-12-00100-f003:**
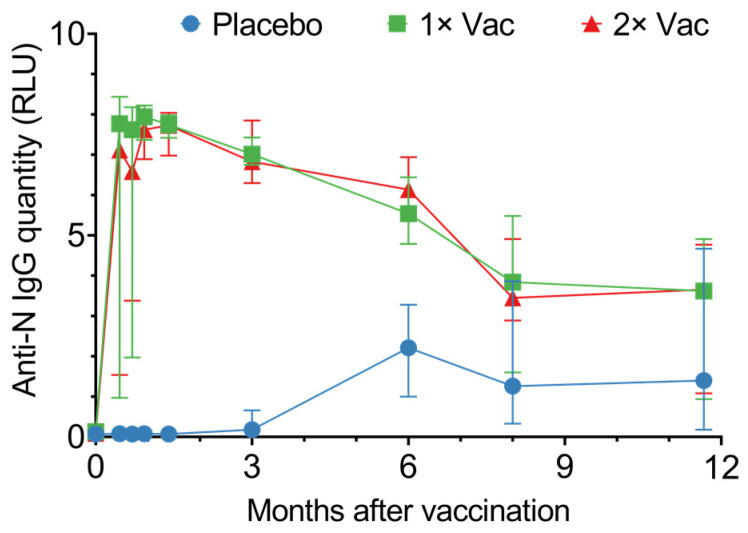
Anti-N IgG quantities in sera of volunteers for the placebo, single (1× Vac), and double (2× Vac) Convacell^®^ vaccination groups. Points represent median values; error bars represent 95% confidence intervals. RLU is relative light units used by the assay. N = 44 for each group. Vaccine was administered on day 0 for both vaccinated groups and on day 21 for the double vaccination group.

**Figure 4 vaccines-12-00100-f004:**
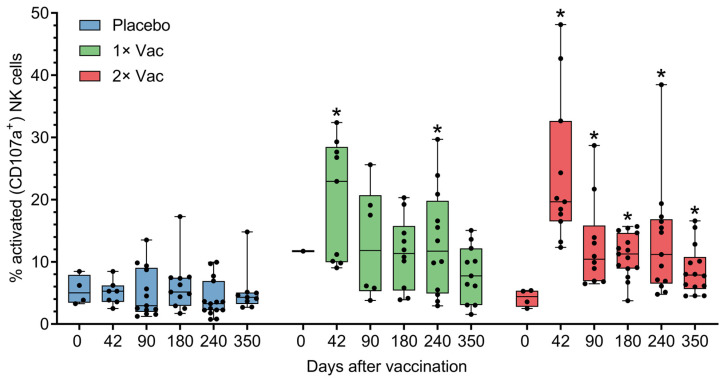
Antibody-dependent natural killer cell activation (ADNKA) in the presence of N-protein antigen by heat-inactivated sera of volunteers in placebo, single (1× Vac), and double (2× Vac) Convacell^®^ vaccination groups, expressed as percentage of CD107a^+^ NK cells in total NK cell population in cultures. Dots represent individual values, overall range is indicated by whiskers, interquartile range is indicated by boxes. Statistically significant according to Mann–Whitney test (i.e., adjusted *p* < 0.05) differences between the value of a vaccinated group and the value of the placebo group at a certain experiment day are indicated with asterisks (*) above the vaccinated group value at corresponding experiment day. Only one replicate in the single vaccination group at day 0 precluded statistical group comparison at this data point. Some data points were excluded from consideration due to either errors in the NK activation assay procedure, or technical limitations precluding sample collection from volunteers. At no experiment day did the values of the two vaccinated groups significantly differ.

**Figure 5 vaccines-12-00100-f005:**
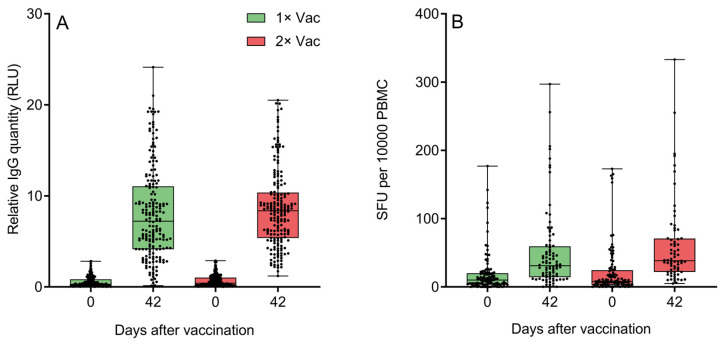
(**A**) Specific anti-N IgG quantification by a chemiluminescent immunoassay on experiment days 0 and 42 in the sera of volunteers (*n* = 438) who received either one or two doses of Convacell^®^. Boxes represent interquartile range; lines are at medians. RLU is relative light units used by the assay. According to Wilcoxon’s signed-rank test, *p* < 0.0001 between day 0 and 42 values in each group. Between groups, according to Mann–Whitney test, *p* = 0.03 on day 0, but *p* = 0.06 on day 42. (**B**) Specific IFNγ-producing PBMC quantification via ELISPOT on experiment days 0 and 42 in the blood of volunteers who received either one or two doses of Convacell^®^. Boxes represent interquartile range; lines are at medians. SFU is spot-forming units used by the assay. According to Wilcoxon’s signed-rank 9 test, *p* < 0.0001 between day 0 and 42 values in each group. Between groups, according to Mann–Whitney test, *p* > 0.99 on day 0 and *p* = 0.07 on day 42.

**Figure 6 vaccines-12-00100-f006:**
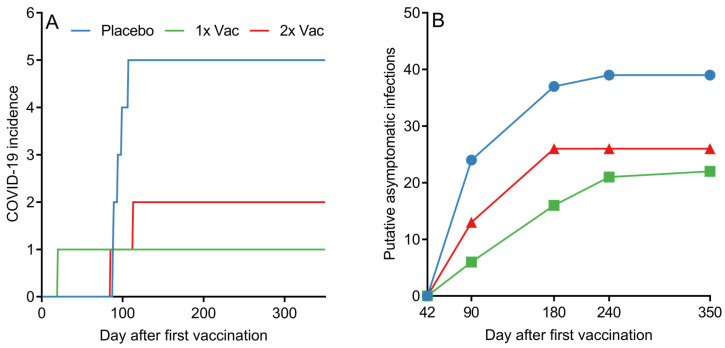
(**A**) Cumulative COVID-19 incidence by day after first vaccination among volunteers in placebo, single (1× Vac) and double (2× Vac) Convacell^®^ vaccination groups in phase II. (**B**) Cumulative putative asymptomatic SARS-CoV-2 infections according to serological data by group past experiment day 42. Value at each time point represents the total number of volunteers who have shown increase in anti-N IgG levels after day 42.

**Table 1 vaccines-12-00100-t001:** Distribution of screened/randomized volunteers be screening center in phase II.

Center #	Location	Volunteers Screened (%)	Volunteers Randomized (%)
1	Saint-Petersburg	31 (18.2%)	25 (18.7%)
2	Saint-Petersburg	102 (60%)	91 (68%)
3	Moscow	16 (9.4%)	12 (9%)
4	Krasnogorsk	21 (12.4%)	6 (4.5%)

**Table 2 vaccines-12-00100-t002:** Volunteer distribution divided into cohort and the number of dropouts by category, for each group in phase II.

	3—Double Vaccination	4—Single Vaccination	5—Placebo
Volunteers included into study (%)	44 (100%)	45 (100%)	45 (%)
Dropped out due to consent withdrawal (%)	0 (0%)	1 (2.2%)	3 (6.7%)
Dropped out due to refusal to cooperate (%)	0 (0%)	1 (2.2%)	1 (2.2%)
Dropped out for other reason (%)	1 (2.3%)	1 (2.2%)	1 (2.2%)
Volunteers included in safety and immunogenicity assessment (%)	44 (100%)	45 (100%)	45 (100%)
Volunteers included in specific post-vaccination immunity investigation cohort—cohort A (%)	15 (34%)	15 (33.3%)	15 (33.3%)
Volunteers included in long-term immunogenicity assessment cohort—cohort B (%)	16 (36%)	15 (33.3%)	15 (33.3%)

**Table 3 vaccines-12-00100-t003:** Volunteer distribution into group and the number of dropouts by category, for each group in phase IIb.

	1—Single Vaccination	2—Double Vaccination
Screened	470 (100%)	
Screened and not included	37 (7.9%)	
Randomized	215 (45.7%)	218 (46.4%)
Dropped out, total	3 (0.7%)	6 (2.8%)
Dropped out due to requiring medical intervention or treatment not allowed by the protocol	1 (0.2%)	0 (0%)
Dropped out due to volunteer violating the protocol of the study	2 (0.5%)	2 (0.9%)
Dropped out due to consent withdrawal	0 (0%)	4 (1.8%)
Included in the immunogenicity assessment study	215 (45.7%)	218 (46.4%)

**Table 4 vaccines-12-00100-t004:** Number and % (in brackets) of volunteers experiencing at least one vaccination-related AE for groups 1 and 2 in phase I safety assessment, for each AE type or system affected. Medical Dictionary for Regulatory Activities (MedDRA) codes are provided for each category and AE.

		Groups 1 + 2 (*n* = 20)	
MedDRA SOC	MedDRA PT	Mild AEs	Moderate AEs
10018065 General disorders and administration site conditions	10022086 Injection site pain	10 (50%)	0
	10022061 Injection site erythema	1 (5%)	0
	10022075 Injection site induration	1 (5%)	0
	10022093 Injection site pruritus	1 (5%)	0
10022891 Investigations	10049187 Red blood cell sedimentation rate increased	1 (5%)	0
	10025258 Lymphocyte count increased	2 (10%)	0
	10047943 Leukocyte count increased	1 (5%)	0
	10003481 Aspartate aminotransferase increased	1 (5%)	0
	10019301 Heart rate decreased	2 (10%)	0
	10029366 Neutrophil count decreased	0	1 (5%)
	10047942 Leukocyte count decreased	1 (5%)	0
	10005758 Blood pressure systolic decreased	1 (5%)	0
	10019303 Heart rate increased	1 (5%)	0
	10005750 Blood pressure increased	1 (5%)	0
	10005760 Blood pressure systolic increased	1 (5%)	0

**Table 5 vaccines-12-00100-t005:** Number and % (in brackets) of volunteers experiencing at least one vaccination-related AE for groups 3, 4, and 5 in phase II safety assessment, for each AE type or system affected. Medical Dictionary for Regulatory Activities (MedDRA) codes are provided for each category and AE.

		Group 3, Double Vaccination (*n* = 44)		Group 4, Single Vaccination (*n* = 45)		Group 5, Placebo (*n* = 45)	
MedDRA SOC	MedDRA PT	Mild AEs	Moderate AEs	Mild AEs	Moderate AEs	Mild AEs	Moderate AEs
10013993 Ear and labyrinth disorders	10014020 Ear pain	1 (2.27%)	0	0	0	0	0
10015919 Eye disorders	10013774 Dry eye	1 (2.27%)	0	0	0	0	0
10017947 Gastrointestinal disorders	10047700 Vomiting	0	1 (2.27%)	0	0	0	0
10018065 General disorders and administration site conditions	10020843 Hyperthermia	2 (4.55%)	0	1 (2.22%)	0	0	0
	10022086 Injection site pain	32 (72.73%)	5 (11.36%)	29 (64.44%)	6 (13.33%)	8 (17.78%)	4 (8.89%)
	10022093 Injection site pruritus	7 (15.91%)	0	4 (8.89%)	1 (2.22%)	0	0
	10022075 Injection site induration	9 (20.45%)	2 (4.55%)	12 (26.67%)	2 (4.44%)	5 (11.11%)	0
	10022004 Influenza-like illness	3 (6.82%)	0	4 (8.89%)	1 (2.22%)	3 (6.67%)	1 (2.22%)
	10022085 Injection site oedema	2 (4.55%)	2 (4.55%)	5 (11.11%)	2 (4.44%)	0	0
	10008531 Chills	1 (2.27%)	1 (2.27%)	1 (2.22%)	0	0	0
	10022061 Injection site erythema	3 (6.82%)	1 (2.27%)	7 (15.56%)	4 (8.89%)	1 (2.22%)	0
	10037660 Pyrexia	4 (9.09%)	0	1 (2.22%)	0	2 (4.44%)	0
	10025482 Malaise	2 (4.55%)	1 (2.27%)	1 (2.22%)	1 (2.22%)		0
	10016256 Fatigue	1 (2.27%)	0	2 (4.44%)	0	1 (2.22%)	0
	10075107 Haemodynamic oedema	0	0	1 (2.22%)	0	0	0
	10003549 Asthenia	0	0	1 (2.22%)	0	1 (2.22%)	0
	10061458 Feeling of body temperature change	0	0	1 (2.22%)	0	0	0
10028395 Musculoskeletal and connective tissue disorders	10028411 Myalgia	2 (4.55%)	1 (2.27%)	1 (2.22%)	1 (2.22%)	0	0
	10003239 Arthralgia	2 (4.55%)	0	0	0	0	0
10029205 Nervous system disorders	10019211 Headache	2 (4.55%)	2 (4.55%)	2 (4.55%)	0	3 (6.67%)	1 (2.22%)
10037175 Psychiatric disorders	10010305 Confusion state	0	0	1 (2.22%)	0	0	0
10038738 Respiratory, thoracic and mediastinal disorders	10043521 Throat irritation	1 (2.27%)	0	0	0	0	0
10040785 Skin and subcutaneous tissue disorders	10020642 Hyperhidrosis	1 (2.27%)	1 (2.27%)	1 (2.22%)	0	0	0
	10037844 Rash	0	0	1 (2.22%)	0	0	0

**Table 6 vaccines-12-00100-t006:** Number and % (in brackets) of volunteers experiencing at least one vaccination-related AE for groups 1 and 2 in phase IIb safety assessment, for each AE type or system affected. Medical Dictionary for Regulatory Activities (MedDRA) codes are provided for each category and AE.

		Group 1, Single Vaccination (*n* = 215)		Group 2, Double Vaccination (*n* = 218)	
MedDRA SOC	MedDRA PT	Mild AEs	Moderate AEs	Mild AEs	Moderate AEs
10005329 Blood and lymphatic system disorders	10025197 Lymphadenopathy	0	0	3 (1.38%)	0
10015919 Eye disorders	10034960 Photophobia	0	0	1 (0.45%)	0
	10003552 Asthenopia	0	0	1 (0.45%)	0
10017947 Gastrointestinal disorders	10012735 Diarrhoea	3 (1.39%)	0	0	0
	10028813 Nausea	1 (0.47%)	0	2 (0.92%)	0
	10000081 Abdominal pain	0	0	2 (0.92%)	1 (0.45%)
10018065 General disorders and administration site conditions	10022075 Injection site induration	15 (6.98%)	2 (0.93%)	21 (9.63%)	5 (2.29%)
	10022086 Injection site pain	68 (31.63%)	5 (2.33%)	74 (33.94%)	13 (5.96%)
	10008531 Chills	9 (4.19%)	2 (0.93%)	7 (3.21%)	9 (4.13%)
	10022093 Injection site pruritus	15 (6.98%)	0	14 (6.42%)	0
	10025482 Malaise		0	10 (4.59%)	4 (1.83%)
	10022061 Injection site erythema	23 (10.7%)	1 (0.47%)	24 (11.01%)	0
	10003549 Asthenia	2 (0.93%)	0	5 (2.29%)	0
	10022085 Injection site oedema	8 (3.72%)	2 (0.93%)	13 (5.96%)	2 (0.92%)
	10037660 Pyrexia	5 (2.33%)	2 (0.93%)	5 (2.29%)	1 (0.45%)
	10022066 Injection site hematoma	1 (0.47%)	0	0	0
	10049438 General physical health deterioration	0	0	0	0
	10025482 Malaise	9 (4.19%)	4 (1.86%)	10 (4.59%)	4 (1.83%)
	10054266 Injection site discomfort	0	0	1 (0.45%)	0
10021881 Infections and infestations	10062106 Respiratory tract viral infection	0	1 (0.47%)	1 (0.45%)	0
10027433 Metabolism and nutrition disorders	10061428 Decreased appetite	1 (0.47%)	1 (0.47%)	3 (1.38%)	0
10028395 Musculoskeletal and connective tissue disorders	10028372 Muscular weakness	4 (1.86%)	2 (0.93%)	7 (3.21%)	2 (0.92%)
	10003239 Arthralgia	2 (0.93%)		6 (2.75%)	1 (0.45%)
	10028411 Myalgia	3 (1.39%)	1 (0.47%)	3 (1.38%)	4 (1.83%)
	10028334 Muscle spasms	0	0	2 (0.92%)	0
10029205 Nervous system disorders	10013573 Dizziness	2 (0.93%)	0	5 (2.29%)	2 (0.92%)
	10019211 Headache	7 (3.26%)	6 (2.79%)	20 (9.17%)	6 (2.75%)
10037175 Psychiatric disorders	10022437 Insomnia	6 (2.79%)	0	5 (2.29%)	1 (0.45%)
	10024642 Listlessness	0	0	1 (0.45%)	0
	10028735 Nasal congestion	0	0	1 (0.45%)	0
	10039101 Rhinorrhoea	0	0	2 (0.92%)	1 (0.45%)
10038738 Respiratory, thoracic and mediastinal disorders	10068319 Oropharyngeal pain	1 (0.47%)	0	0	2 (0.92%)
	10011224 Cough	0	0	0	2 (0.92%)
10040785 Skin and subcutaneous tissue disorders	10020642 Hyperhidrosis	5 (2.33%)	2 (0.93%)	5 (2.29%)	2 (0.92%)
10047065 Vascular disorders	10020772 Hypertension	0	1 (0.47%)	0	0

## Data Availability

The data presented in this study are available on request from the corresponding author.

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
