# Peer review of "Safety and Immunogenicity of the Convacell® Recombinant N Protein COVID-19 Vaccine"

_vaccines, 2024, doi:10.3390/vaccines12010100_

Round 1

Reviewer 1 Report

Comments and Suggestions for Authors

Estimated Authors,

I've been invited to review this interesting report on the Phase I+II+IIb study on the new putative CONVACELL(R) vaccine.

The main novelty of this formulate insists in its target, the N protein instead of the S(pike) protein, that is usually targeted by available vaccines (mRNA and non-mRNA based vaccines). Convacell, a putative vaccine obtained through recombinant technology, has many potential "plus" that represent the main interest of this study (i.e. the higher stability of N protein compared to S protein, with a likely reduced risk for evolutive escape from immunitary pressure from available vaccines, a more conventional technology, and less potential claims for side effects and potential complications associated with mRNA techology). However, the main shortcoming of this study remains largely unanswered across introduction and discussion: what is the rationale for a vaccine targeting N protein? Authors should explain whether such a target could have any significance in hindering the pathogenicity of SARS-CoV-2.

Regarding the design of the paper, I think that some sections should be removed, and more precisely the sections of the main text dealing with the putative efficacy of vaccine (i.e. chapter 3.6 and corresponding section within discussion). The study was not designed for assessing potential efficacy of this formulate, and even though Authors are quite clear in discussing how cautiously their available data should be ascertained, their reporting may be confusing. A follow up study on this topic will be very interesting.

Another improvement would be represented by some minor amendments of Table 1, 2, 3 by including per cent values.

Author Response

We would like to kindly thank the esteemed reviewer for taking their time to review our work and provide meaningful avenues for improvement.

Our rationale for a nucleocapsid-based COVID-19 vaccine is that the N-protein is supported by multiple studies that have demonstrated that nucleocapsid-based COVID-19 vaccines are highly immunogenic and able to protect against severe disease and generate long-lasting immunity (1–7). Furthermore, there are studies describing that viral nucleocapsid proteins in general are good targets for vaccines (8–10). We have amended the text of the article to include the abovementioned references and information (please see Introduction section).

We would like to express our wish for the retaining of section 3.6. We believe information about PCR-confirmed cases represents an interesting piece of data exploration and can be used to illustrate the trend that we are expecting to observe in Convacell®’s ongoing phase III clinical trials. Continuous anti-N IgG level monitoring is not included in the protocol of phase III efficacy study and thus putative virus encounter rate derived from phase II data in our article is an interesting addition to our overall research. Ample warnings have been made to the reader regarding the statistical significance of our findings due to small sample size in phase II. To not coufuse reader about epidemiological efficacy of vaccine we have renamed the section “3.6. Putative effectiveness” to “3.6. Infections and virus encounters”.

We have amended the tables 1, 2 and 3 in accordance with your commentary.

Works cited: 

  1. Bai Z, Cao Y, Liu W, Li J. The SARS-CoV-2 Nucleocapsid Protein and Its Role in Viral Structure, Biological Functions, and a Potential Target for Drug or Vaccine Mitigation. Viruses. 2021 Jun 10;13(6):1115.
  2. Thura M, Sng JXE, Ang KH, Li J, Gupta A, Hong JM, et al. Targeting intra-viral conserved nucleocapsid (N) proteins as novel vaccines against SARS-CoVs. Bioscience Reports. 2021 Sep 30;41(9):BSR20211491.
  3. Dutta NK, Mazumdar K, Gordy JT. The Nucleocapsid Protein of SARS–CoV-2: a Target for Vaccine Development. Dutch RE, editor. J Virol. 2020 Jun 16;94(13):e00647-20, /jvi/94/13/JVI.00647-20.atom.
  4. Tilocca B, Soggiu A, Sanguinetti M, Musella V, Britti D, Bonizzi L, et al. Comparative computational analysis of SARS-CoV-2 nucleocapsid protein epitopes in taxonomically related coronaviruses. Microbes and Infection. 2020 May;22(4–5):188–94.
  5. Sieling P, King T, Wong R, Nguyen A, Wnuk K, Gabitzsch E, et al. Prime hAd5 Spike + Nucleocapsid Vaccination Induces Ten-Fold Increases in Mean T-Cell Responses in Phase 1 Subjects that are Sustained Against Spike Variants [Internet]. Allergy and Immunology; 2021 Apr [cited 2022 Nov 14]. Available from: http://medrxiv.org/lookup/doi/10.1101/2021.04.05.21254940
  6. Matchett WE, Joag V, Stolley JM, Shepherd FK, Quarnstrom CF, Mickelson CK, et al. Cutting Edge: Nucleocapsid Vaccine Elicits Spike-Independent SARS-CoV-2 Protective Immunity. JI. 2021 Jul 15;207(2):376–9.
  7. Van Elslande J, Oyaert M, Ailliet S, Van Ranst M, Lorent N, Vande Weygaerde Y, et al. Longitudinal follow-up of IgG anti-nucleocapsid antibodies in SARS-CoV-2 infected patients up to eight months after infection. Journal of Clinical Virology. 2021 Mar;136:104765.
  8. Gil L, López C, Lazo L, Valdés I, Marcos E, Alonso R, et al. Recombinant nucleocapsid-like particles from dengue-2 virus induce protective CD4+ and CD8+ cells against viral encephalitis in mice. International Immunology. 2009 Oct 1;21(10):1175–83.
  9. Wraith DC, Vessey AE, Askonas BA. Purified Influenza Virus Nucleoprotein Protects Mice from Lethal Infection. Journal of General Virology. 1987 Feb 1;68(2):433–40.
  10. Huang B, Wang W, Li R, Wang X, Jiang T, Qi X, et al. Influenza A virus nucleoprotein derived from Escherichia coli or recombinant vaccinia (Tiantan) virus elicits robust cross-protection in mice. Virol J. 2012 Dec;9(1):322.

Reviewer 2 Report

Comments and Suggestions for Authors

Review of Rabdano et al, “Safety and Immunogenicity of the Convacell® Recombinant N Protein COVID-19 Vaccine”, vaccines-2785632

The group of Veronika Skvortsova present a manuscript describing a combined phase I/II clinical trial for their novel Covid 19-vaccine Convacell.  The vaccine is based upon a recombinant N protein. The group reports no adverse effects were observed and 100% or the participants seroconverted by day 42 with sustained anti-N IgG levels for 350 days with involvement of both the cellular and humoral immune responses. Overall, the paper was very well written and thoroughly presented. I found the presented data to be very inclusive and supportive of the author’s claims that they have a relatively safe and effective N protein-based subunit vaccine for SARS-CoV-2.  

Major

Moderate

Lines 310-314 Were the cases of PCR confirmed SARS-CoV-2 infection excluded from further analyses?

Line 518-522 “Notably, the dose of 50 ug of protein N in Convacell® is much higher than in other currently developed vaccines containing N as an antigen, where the amount of total antigen could be as low as 3 μg (51–53), which further emphasizes the high immunogenicity and promise of Convacell®.”  I’m a little confused by this statement.  Wouldn’t the usage of almost 20x a potential minimal dose be less promising?  Are the authors attempting to state that they could potentially lower the inoculating dosage and still achieve good immune responses?  I think this statement needs clarification.

I think the screening procedure should be elucidated to ensure volunteers were not previously exposed to SARS-CoV-2.

The authors may wish to elaborate on the primary circulating strains of SARS-CoV-2 on which Figure 6 data is based.  Seems like it would be an Omicron variant based upon the dates.  This would suggest cross protection of Wuhan N and this newer variant?

Figure 3 Why is the trend in Anti-N IgG increasing for placebo but decreasing for vaccinates?  Wouldn’t one assume that the group trends should be somewhat consistent?

Figure 6 do the authors have any thoughts as to why double vaccinated patients appear to have both a higher COVID-19 and asymptomatic incidence rate compared to single vaccinates?  Based upon these results could the authors argue that a single vaccine dose would be superior to boosters?

Minor

line 31 “able to both to protect” remove redundant word “to”

Line 40 “conservative and less prone to accumulate mutations” change conservative to “highly conserved” or “genetically stable between strains”

Line 84 is it necessary to mention that 170 people were screened rather than just 135 were utilized?

Line 179 “PMBC” should be “PBMC”

Line 200 define DPBS as Dulbecco’s phosphate buffered saline.

Line 210 “cultural” should be culture

Line 229 extra line space present

Line 231 “takes” should be “taken”

Line 265 “Table 1)

Line 269 “Table 2”

Line 281 “Table 3”

Comments on the Quality of English Language

Mostly well done.  I included a few minor word choice alterations.

Author Response

Esteemed reviewer, we thank you for your consideration and would like to reply to your comments in point by point way.

Lines 310-314 Were the cases of PCR confirmed SARS-CoV-2 infection excluded from further analyses?

Yes, volunteers with COVID-19 were excluded from immunogenicity data presented in the paper after their diagnosis was confirmed.

Line 518-522 “Notably, the dose of 50 ug of protein N in Convacell® is much higher than in other currently developed vaccines containing N as an antigen, where the amount of total antigen could be as low as 3 μg (51–53), which further emphasizes the high immunogenicity and promise of Convacell®.”  I’m a little confused by this statement.  Wouldn’t the usage of almost 20x a potential minimal dose be less promising?  Are the authors attempting to state that they could potentially lower the inoculating dosage and still achieve good immune responses?  I think this statement needs clarification.

The esteemed reviewer is correct that the above passage is confusing. Upon reflection, we have judged it to fulfill no particular need in our paper and as such have excluded it from the revised version of the manuscript.

I think the screening procedure should be elucidated to ensure volunteers were not previously exposed to SARS-CoV-2.

We have not selected for SARS-CoV-2-naïve volunteers due to the technical difficulties of finding such individuals in the ongoing pandemic environment as well as the impossibility of detecting individuals who had past encounters with SARS-CoV-2 but did not develop COVID-19 as a result. To correct for the effects of SARS-CoV-2 encounters and previous immunizations, our experiment in phases II and IIb was designed to also analyze the levels of anti-N IgGs in the sera of volunteers on day 0, before vaccination took place. These data were then used to deduce anti-N IgG increases compared to their initial background levels. We did observe intra-group separation during the immunogenicity experiments that could be explained by previous SARS-CoV-2 encounters – namely, the speed of forming a humoral immune response after vaccination appears to be correlated positively with initial anti-N IgG levels (see Figure S4). However, both subgroups converged at day 48 for 100% seroconversion. Thus, we do not believe that our vaccine is in any way less suitable for SARS-CoV-2-naïve individuals than it is for those who have encountered the virus or were vaccinated previously.

The authors may wish to elaborate on the primary circulating strains of SARS-CoV-2 on which Figure 6 data is based.  Seems like it would be an Omicron variant based upon the dates.  This would suggest cross protection of Wuhan N and this newer variant?

The esteemed reviewer is correct here. The primary circulating strain of the virus at the time of the study was indeed Omicron. The cross-reaction of T-lymphocytes of Convacell® vaccinated individuals to peptides derived from the protein N of the Wuhan, Delta and Omicron variants of SARS-CoV-2 was researched and demonstrated in our earlier paper (DOI: 10.47183/mes.2022.033). We have added circulating strain information to the discussion section.

Figure 3 Why is the trend in Anti-N IgG increasing for placebo but decreasing for vaccinates?  Wouldn’t one assume that the group trends should be somewhat consistent?

Some individuals in the placebo group could be expected to have developed anti-N IgGs due to the spike in SARS-CoV-2 encounters and COVID-19 cases in Russia during the course of the study (please see Fig. S15). The titers of anti-N IgG, meanwhile, could be expected to decrease in vaccinated individuals over time as their immunity deteriorates. As such, we believe that the group trends observed during the study cannot be said to majorly differ from what is expected.

Figure 6 do the authors have any thoughts as to why double vaccinated patients appear to have both a higher COVID-19 and asymptomatic incidence rate compared to single vaccinates?  Based upon these results could the authors argue that a single vaccine dose would be superior to boosters?

Due to the results of phase IIb (n=433) showing that no statistically significant difference in intensity of the immune response exists between the single vaccination and double vaccination group, we believe that the phenomenon to which the esteemed reviewer refers is simply due to small sample size in phase II (n=135). Again due to the results of phase IIb, we do not believe that the 1-dose vaccination regimen for Convacell® is superior to the 2-dose vaccination regimen in any manner aside from the decreased cost and organizational effort of arranging half the vaccination appointments.

Various minor concerns

The authors thank the esteemed reviewer for his comments and have adjusted the text accordingly.